# Exploring Mechanotransduction and Inflammation in Human Cartilaginous Endplate Cells in Blended Collagen–Agarose Hydrogels Under Cyclic Compression

**DOI:** 10.3390/gels11090736

**Published:** 2025-09-12

**Authors:** Katherine B. Crump, Chloé Chapallaz, Ahmad Alminnawi, Paola Bermudez-Lekerika, Liesbet Geris, Jérôme Noailly, Benjamin Gantenbein

**Affiliations:** 1Tissue Engineering for Orthopaedics & Mechanobiology, Bone & Joint Program, Department for BioMedical Research, Faculty of Medicine, University of Bern, 3008 Bern, Switzerland; katherine.crump@unibe.ch (K.B.C.); paola.bermudez@unibe.ch (P.B.-L.); 2Graduate School for Cellular and Biomedical Sciences, University of Bern, 3012 Bern, Switzerland; 3GIGA In Silico Medicine, University of Liège, 4000 Liège, Belgium; ahmad.alminnawi@uliege.be (A.A.); liesbet.geris@kuleuven.be (L.G.); 4Skeletal Biology and Engineering Research Center, KU Leuven, 3000 Leuven, Belgium; 5BCN MedTech, Department of Engineering, Universitat Pompeu Fabra, 08002 Barcelona, Spain; jerome.noailly@upf.edu; 6Department of Orthopedic Surgery & Traumatology, Faculty of Medicine, Inselspital, Bern University Hospital, University of Bern, 3010 Bern, Switzerland

**Keywords:** collagen, hydrogel, mechanobiology, catabolism, intervertebral disc, cartilage

## Abstract

Little is known about cartilaginous endplate (CEP) mechanobiology or how it changes in a catabolic microenvironment, partly due to difficulties in conducting mechanotransduction in vitro. Recent studies have found blended collagen–agarose hydrogels to offer improved mechanotransduction in chondrocytes compared to agarose alone. It was hypothesized that blended collagen–agarose hydrogels would be sufficient to improve the mechanobiological response in CEP cells relative to that in agarose alone, while maintaining the chondrocyte phenotype and ability to respond to pro-inflammatory stimulation. Thus, human CEP cells were seeded into blended 2% agarose and 2 mg/mL type I collagen hydrogels, followed by culture with dynamic compression up to 7% and stimulation with TNF. Results confirmed CEP cells retained a rounded phenotype and high cell viability during culture in blended collagen–agarose hydrogels. Additionally, TNF induced a catabolic response through downregulation of pericellular marker *COL6A1* and anabolic markers *ACAN* and *COL2A1*. No significant changes were seen due to dynamic compression, suggesting addition of collagen to agarose was not sufficient to induce mechanotransduction in human CEP cells in this study. However, blended collagen–agarose hydrogels increased stiffness by 4× and gene expression of key cartilage marker *SOX9* and physioosmotic mechanosensor *TRPV4*, offering an improvement on agarose alone.

## 1. Introduction

Intervertebral disc (IVD) degeneration remains a leading contributor to low back pain, with no currently effective treatment capable of reversing the degenerative process [1]. The IVD is a complex structure that is essential for spinal flexibility and managing mechanical loads. It is made up of three main regions: the highly hydrated nucleus pulposus (NP) at the core, the fibrous annulus fibrosus (AF) laterally surrounding the NP, and the cartilaginous endplates (CEP), which separate the IVD from the vertebral bone [1]. Within a healthy disc, there is a balance between anabolic activity, including the production of collagen and glycosaminoglycans (GAGs), and catabolic activity, such as the production of matrix metalloproteinases (MMPs) involved in extracellular matrix (ECM) turnover. Mechanical loading plays a central role in regulating this balance, but both excessive and insufficient mechanical cues can be problematic [2,3,4]. This damage can accumulate over time, leading to an impaired response to mechanical loading and causing further damage [5]. Further, this damage leads to changes at the cellular level [1,6]. IVD cells have been shown to have different mechanobiological responses in a pro-inflammatory environment in comparison to a healthy environment, suggesting that mechanotransduction pathways are altered in degeneration [7,8,9,10].

The biochemical composition of the CEP plays a key role in determining the permeability and response to mechanical loading, as the CEP must be stiff enough to sustain disc pressure but porous enough to facilitate solute transport [11,12,13]. Mechanical simulations of the disc suggest that initial mechanical failure of the IVD occurs in the endplates, and damage to the CEP has been recognized to play a key role in early IVD degeneration and low back pain [14]. CEP damage can also provoke inflammatory crosstalk between the disc and bone marrow, thereby increasing the concentrations of pro-inflammatory cytokines and chemokines, such as interleukin-1β (IL-1β) or TNF, in the disc microenvironment [15]. Additionally, calcification of the CEP occurs during aging and degeneration, leading to impaired nutrient transport [16,17]. Despite the fundamental role that the CEP plays in mechanics and IVD degeneration, little is known about CEP mechanobiology, particularly how it is altered within degenerative environments. This is in part due to the difficulty in conducting in vitro mechanotransduction studies, which require advanced 3D culture systems [18].

Agarose has been considered the gold standard in biomaterials for 3D culture of cartilaginous tissues, because it is biocompatible, preserves a round cell morphology, and is strong enough to remain stable under compressive loading [19,20,21]. However, agarose lacks cell adhesion motifs and is bioinert, preventing mechanosensors, such as integrins, from interacting with the ECM and initiating the mechanotransduction cascade [19]. A pre-culture period has been shown to produce a pericellular matrix (PCM) that can facilitate the transmission of mechanical signals to the cells [22]. However, prior studies in our lab have found that dynamic compression of human CEP cells in 2% agarose did not produce changes at the transcriptomic or proteomic level [18]. Consequently, several studies have blended agarose with other biomaterials that support ECM–cell interactions such as collagen [19,23,24], laminin [24], methacrylated gelatin [25], and hyaluronic acid [26]. Further, biomimetic hydrogels that mimic the native ECM have been developed for different cartilaginous tissues. Hydrogels formed from decellularized ECM derived from mesenchymal stem cells have been demonstrated to form hyaline cartilage-like tissue after injection in vivo [27]. Jacobs et al. (2023) also developed a biomimetic IVD that mimics the poroelastic and viscoelastic behavior of a natural disc under loading, consisting of a hydrogel made of hydroxyethyl-methacrylate and sodium-methacrylate representing the NP, a fibrous jacket for the AF, and titanium plates representing the endplates [28]. Other studies have also fabricated a biomimetic hydrogel that allows successful diffusion of glucose and lactic acid to represent the CEP through crosslinking hyaluronic acid, chondroitin sulfate, and type II collagen [29].

Type I collagen is among the most commonly used scaffolds for 3D culture and tissue engineering applications and the most prevalent ECM protein in mammalian tissues [23]. Ulrich et al. adjusted the agarose content in collagen–agarose scaffolds to develop a straightforward and cost-effective biomaterial platform that allows tuning of ECM’s biophysical properties independently of collagen content in a 3D environment [23]. Cambria et al. developed blended agarose–collagen hydrogels for culture of NP cells, finding the formation of a homogeneous, interconnected network that maintained both the mechanical strength of agarose and the biofunctionality of type I collagen [19]. Introducing collagen to agarose hydrogels increased cell adhesion and focal adhesion kinase (FAK) activation while preserving the low proliferation and rounded morphology of chondrocyte-like cells [19]. Thus, we hypothesized that 3D culture in blended collagen–agarose hydrogels would improve mechanotransduction of CEP cells in comparison to culture in agarose alone. This study aimed to recreate the experiment in Crump et al. using blended collagen–agarose hydrogels in place of 2% agarose to investigate the effects of dynamic compression at 7% strain and stimulation with 10 ng/mL TNF in human CEP cells.

## 2. Results and Discussion

### 2.1. Blended Collagen–Agarose Hydrogels Are Stiffer than Agarose

Material stiffness was determined in the blended 2% and 2 mg/mL type I collagen compared to 2% agarose, the gold standard for 3D culture of chondrocytes. The blended collagen–agarose hydrogels were found to be 4× stiffer than agarose hydrogels, with a calculated Young’s modulus (E) of 15.53 kPa and 3.74 kPa, respectively (Figure 1A). Additionally, both hydrogels maintained a linear, elastic region up to 10% strain.

### 2.2. Cell Viability Was Not Affected by TNF Treatment or Dynamic Compression

To evaluate whether the blended hydrogel formulation of 2% agarose with 2 mg/mL type I collagen could support CEP cells in 3D culture, we assessed live/dead cell distributions at the hydrogel center and periphery on days 0, 7, and 14 (Figure 1B). At baseline, cell viability ranged from 51.0 to 99.0%, with medians of 60.3% and 59.1% in the center and edge, respectively. Across all conditions, neither TNF stimulation nor cyclic loading significantly altered viability over the 14-day culture period. Median values remained within 69.7–86.9% throughout, although donor-to-donor variability increased after culture. In unloaded controls, viability spanned 57.8–90.9%, while in dynamically compressed controls it was within the same range (57.8–90.9%). For TNF-treated samples, values were 57.7–92.8% without loading and 46.2–91.5% under dynamic compression.

### 2.3. Cell Height Was Significantly Decreased During Culture

The heights of collagen–agarose hydrogels were measured on days 0, 7, and 14 (Figure 1C). While there was no significant difference between conditions, the hydrogel height significantly decreased from ~3 mm to ~2.5 mm in all conditions after 7 days. This height decrease happened within the first week, and then height was maintained at ~ 2.5 mm for the second week in all conditions. It should be noted that only two biological replicates were included for height measurement of unloaded hydrogels at day 14; therefore, statistics could not be performed. However, values followed the same trend, maintaining a height of ~2.5 mm.

### 2.4. Human CEP Cells Retained Their Rounded Morphology 

Human CEP cells in unloaded control hydrogels maintained their rounded morphology in the blended collagen–agarose hydrogels throughout the experiment, as demonstrated by F-actin staining with phalloidin (Figure 2A). The actin cytoskeleton was enhanced after 7 and 14 days in comparison to the initial intensity at day 0 for all conditions (Figure 2B). Notably, unloaded hydrogels showed a higher phalloidin intensity relative to the number of cells in comparison to dynamically loaded hydrogels. While TNF did not appear to influence the intensity of actin staining, elongated fibers were found in several locations within the hydrogel. These elongated fibers were also present in the dynamically compressed control hydrogels but not in unloaded control hydrogels. However, a majority of cells retained their rounded morphology in all conditions.

### 2.5. Cell Metabolism and GAG/DNA Content Remained Constant

Cell metabolism and GAG production did not show significant differences after 7 or 14 days of culture for any condition. However, GAG content showed an insignificant increase from day 0 (14.5 μg) relative to day 7 (18.3–21.0 μg) and day 14 (19.2–23.9 μg) in all conditions (Figure 3A). This trend disappears at day 7 when GAG content is normalized to DNA, but at day 14 normalized GAG content is insignificantly increased 1.2–2.5× initial values (Figure 3B). This pattern may be partly due to an insignificant increase in DNA content in some donors after 7 days, which returns to the initial DNA content values after 14 days. Cell metabolic activity was also unaffected after 7 days, but at day 14, it was insignificantly increased by 1.6–2.4 times in comparison to the initial cell metabolic activity at day 0. DNA content was not significantly different between groups; however the median increased between 1.35 and 1.95× after 7 days relative to day 0 (Figure 3C). However, DNA content returned to a median of a 0.72–1.27× fold change from day 0 after 14 days culture, suggesting early cell proliferation before returning near to initial cell counts.

### 2.6. Nitric Oxide (NO) Content

NO content was not significantly different between culture conditions (Figure 3E). However, TNF-treated, unloaded hydrogels saw significantly higher NO release into the media after 7 days in comparison to 14 days (*p* = 0.008).

### 2.7. Anabolic Gene Exression Decreases in Response to TNF Stimulation

To investigate the response of CEP cells to TNF stimulation and dynamic compression, the expression of anabolic, catabolic, and mechanobiological genes was quantified using qPCR (Figure 4). Dynamic compression and culture duration did not significantly alter gene expression, whereas treatment with 10 ng/mL TNF significantly affected the expression of several markers, including aggrecan (*ACAN*), type II collagen (*COL2A1*), and type VI collagen (*COL6A1*).

After 7 days of culture, TNF-treated constructs exhibited reduced expression of anabolic genes. Specifically, dynamically loaded hydrogels showed significantly less *ACAN* (*p* = 0.048) and *COL2A1* (*p* = 0.049) expression after 7 days when stimulated with TNF compared to controls. Additionally, *ACAN* expression trended towards a decrease in TNF-treated unloaded and dynamically compressed hydrogels after 14 days (*p* = 0.064). TNF stimulation significantly decreased PCM marker *COL6A1* expression in both unloaded and dynamically compressed hydrogels after 7 and 14 days (*p* < 0.04).

*IL-6* expression showed an insignificant increase in response to TNF stimulation but varied greatly between donors. Similarly, *MMP-3* increased in each condition relative to day 0; however, it was not affected by TNF stimulation or dynamic compression. No effects were seen in *COL1A2*, *COL10A1*, SRY-Box Transcription Factor 9 (*SOX9)*, A disintegrin and metalloproteinase with thrombospondin motifs 5 *(ADAMTS5)*, or *MMP-13* expression. Integrin subunits α5 and β1 were not significantly affected by TNF stimulation or dynamic compression, except for a trend in a decrease in *ITGB1* in dynamically loaded hydrogels stimulated with TNF relative to controls (*p* = 0.097). Transient receptor potential cation channel subfamily V member 4 (*TRPV4)* expression was also unaffected by either TNF stimulation or mechanical loading.

Transcript-level changes were similar to those previously reported in dynamic compression and TNF stimulation of human CEP cells cultured in 2% *w*/*v* agarose [18]. However, culture in blended collagen–agarose hydrogels significantly upregulated expression of *SOX9* (*p* = 0.017–0.021) for all conditions and timepoints and *TRPV4* relative to culture in only agarose (*p* = 0.011–0.032), except in unloaded TNF-stimulated hydrogels at day 7 (*p* = 0.066) and control hydrogels at day 14 (*p* = 0.129).

### 2.8. Reaction Forces of the Finite Element Model

Finite element (FE) simulations of differing hydrogel heights predicted maximum reaction forces within the range of experimentally measured reaction forces (Figure 5A). The experimentally measured reaction forces were calculated as the maximum reaction force measured within 10–20 min in the one hour of dynamic compression and then divided by 6 to account for the number of hydrogels in each chamber, and they were found to be 0.1052 ± 0.03378 N. The FE model simulating dynamic compression of 3 mm hydrogels predicted a maximum reaction force of 0.08226 N, corresponding to the minimum of the experimental range. In contrast, simulations of the 2 mm hydrogels predicted a maximum reaction force of 0.1358 N, corresponding to the maximum value within the experimental range. Simulations of the 2.5 mm hydrogel, which was found to be the average height of hydrogels after 7 and 14 days of culture, predicted a maximum reaction force of 0.1024 N, consistent with the average experimentally measured maximum reaction forces.

### 2.9. Fluid Exchange Between the Models and the Surrounding Environment

Reactive fluid volume flux (RVT) simulations predicted that during compression, fluid was expulsed from the hydrogel, and fluid was reabsorbed upon release. Notably, predictions of reaction total fluid volume (RVF) suggested that the hydrogel did not fully reabsorb the expulsed fluid before the start of the subsequent compression phase (Figure 5B). One minute of simulated loading predicted a cumulative fluid volume loss of 1.1 mm^3^.

### 2.10. Discussion

While blended collagen–agarose hydrogels offered promising properties for permitting mechanotransduction in cartilaginous cells, dynamic compression of human CEP cells seeded into a 2% *w*/*v* agarose and 2 mg/mL collagen hydrogel did not lead to any significant changes in the cellular response. Thus, similar to agarose alone, it is not an ideal hydrogel to evaluate mechanobiology in human CEP cells. However, the 3D culture of CEP cells in blended collagen–agarose hydrogel was sufficient to examine the effects of cytokine stimulation on cellular response. At the transcript level, TNF stimulation induced downregulation of anabolic and pericellular markers *ACAN*, *COL1A2*, and *COL6A1*, suggesting that TNF induces the breakdown of the ECM and hinders the formation of a PCM.

Median cell viability for each condition (between 69.7 and 86.9%) suggests high biocompatibility, although one donor exhibited lower cell viability below 50% after stimulation with TNF and dynamic compression. Additionally, cell proliferation occurred in the first 7 days of culture; however, it returned to initial numbers after 14 days, suggesting a time-dependent response to culture in blended collagen–agarose hydrogels. NO release into the media was also time-dependent, and it increased following TNF stimulation after one week but then returned to initial levels after two weeks’ culture. Despite no significant differences between cell viability and GAG content relative to initial values in any condition, hydrogel height was significantly decreased after 7 and 14 days. Moreover, cell height decreased approximately 0.5 mm in conditions after one week. Height did not decrease further after the second week, indicating that height changes occur early in culture and stabilize over time. Interestingly, the loss in height was measured to be the same in unloaded and dynamically compressed hydrogels, indicating the elasticity of the hydrogel. This height loss in the unloaded condition could be due to the weight of the chamber lids, which exerted ~5.1 Pa per hydrogel, impeding a true free swelling state. Nevertheless, the absence of any further height loss in dynamically loaded hydrogels suggests that the blended collagen–agarose hydrogels are able to retain their structure following strain deformations.

The varied height also affected the load each hydrogel experienced. The bioreactor delivered a set displacement to all six hydrogels within the chamber, and only the total applied force was measured. Thus, it was not possible to determine the load per hydrogel. This likely played a role in the variance in reaction force measured during the experiment, and this was supported by the corresponding FE analysis. As the hydrogel height decreased, the strain increased (if the hydrogel height were only 2 mm, the strain would have reached 10.5%), and the predicted reaction force following compression increased. Simulations of hydrogels with heights between 2 and 3 mm showed a maximum reaction force within the range of experimentally measured forces. Consequently, it is likely that the height differences play a significant role in the load experienced by each hydrogel and could be a potential source of error when comparing hydrogels even within the same chamber.

Additionally, simulated fluid dynamics predicted that fluid lost during compression is not fully recovered, leading to changes in volume over time. Thus, it is possible fluid loss could contribute to the decreased height seen in the hydrogel experiment. As no significant differences were observed between unloaded and dynamically compressed hydrogels at the transcript or biochemical level, this variation does not appear to have a strong influence in this experiment. However, future experiments would need to quantify fluid volume before and after loading to confirm that fluid loss contributed to the hydrogel height loss.

The addition of collagen to agarose also made a much stiffer hydrogel in comparison to standard 2% agarose, increasing the Young’s modulus by a 4× fold change. Further, stiffness likely increases with culture time due to the cellular deposition of ECM components [21]. Regrettably, this still does not reach the stiffness of native CEP tissue, although the stiffness was close to the reported PCM stiffness of a chondron, which ranges from 17 to 200 kPa [30]. Lower stiffness values are a closer match to osteoarthritic chondrons [30]; however, this has not been measured in the CEP, where calcification could lead to a stiffer environment [17]. Thus, future 3D culture of CEP cells should consider a stiffer hydrogel to better represent native CEP. Additionally, the blended hydrogels enabled a majority of cells to preserve their rounded morphology, indicative of a healthy cartilaginous phenotype. This rounded morphology is also observed in agarose, whereas cells in pure collagen hydrogels typically exhibit an elongated morphology reminiscent of a fibroblast phenotype [19]. Notably, TNF treatment led to the formation of some stretched fibers after 14 days of culture. This pattern was not seen in the unloaded controls; however, it was present in dynamically loaded controls. This suggests that damage is incurred from both TNF stimulation and dynamic compression, resulting in locations that display fibrotic characteristics. At this level of dynamic compression, damage should not occur, as it is within the physiological range of <20% [31,32]. However, with the lack of cellular response to dynamic compression seen at the transcript and protein level, it is possible that a 7% loading at 1.5 Hz was not sufficient to induce changes to the cells, and a greater load would be necessary. Previous studies have used higher strains of 10–20% to induce chondrocyte mechano-activation; however, significant changes have been seen in chondrocytes following as low as a 4.5% strain. Notably, at higher loads, cracking was observed in the hydrogels; thus, another possibility to induce mechanotransduction would be to increase the amount of time per day that the hydrogels are loaded, exceeding one hour.

At the transcript level, cellular response to TNF was similar between CEP cells seeded into blended collagen–agarose and cells seeded into agarose alone. In contrast, the gene expression of *MMP-3* after 7 days of culture differed; it was elevated in TNF-stimulated agarose hydrogels relative to controls but not in blended collagen–agarose hydrogels. Notably, the expression of transcription factor *SOX9* and calcium channel *TRPV4* were significantly increased with the addition of collagen to the hydrogels in comparison to expression in agarose alone. TRPV4 has been demonstrated to play a key role in mechanotransduction in chondrocyte signaling and regulate ECM biosynthesis [33,34]. Further, *TRPV4* has been previously shown to upregulate activity of *SOX9*, essential for chondrogenesis [34]. Thus, the upregulation of both *TRPV4* and *SOX9* suggests that blended collagen–agarose hydrogels offer a more anabolic environment for CEP cells than agarose.

However, the downregulation of anabolic and pericellular markers *ACAN*, *COL2A1*, and *COL6A1* in response to TNF stimulation was consistent with prior reported results in agarose [18]. Nonetheless, at the protein level, GAG production remained unchanged relative to day 0 for all conditions. Likewise, this is consistent with the results of GAG content in 2% agarose hydrogels. Thus, blended collagen–agarose hydrogels did not offer improved mechanotransduction in this study but remain a viable hydrogel for investigating cytokine stimulation and microenvironment perturbations in 3D culture of human CEP cells, with a stiffness closer to that of native CEP tissue.

## 3. Conclusions and Limitations

In summary, blended collagen–agarose hydrogels provided a suitable platform to investigate the response of human CEP cells to pro-inflammatory stimulation. TNF exposure elicited a catabolic shift, reflected by reduced expression of anabolic genes, consistent with previous findings in agarose-only cultures. In contrast, dynamic compression at 7% strain did not alter gene expression in these constructs, suggesting that future mechanotransduction studies should consider modifying the loading regimen by increasing the strain or employing alternative hydrogels that better support integrin-mediated adhesion and signaling.

Several limitations of the study should be acknowledged. Mechanosensor activation was only evaluated at the transcript level. Protein-level confirmation of mechanosensor inactivation or phosphorylation should be evaluated through immunohistochemistry or Western blot. The comparison to human CEP cells cultured in 2% *w*/*v* agarose had several differences in the study design, including the timing of collecting the hydrogels following loading, the use of low-glucose (1 g/L) DMEM, and the use of unmatched donors. Additionally, the tissue used to extract CEP cells came from trauma patients undergoing spinal surgery, which was considered a healthy phenotype. The patients were healthy, with no history of disc degeneration; however, the trauma spinal injury itself could have influenced the cells. The FEM also had several limitations. Material properties that were not available for blended collagen–agarose hydrogels were approximated from studies on 2-3% agarose hydrogels. Moreover, the fluid flow was not experimentally validated; therefore, the interpretation that height loss may result from fluid loss remains theoretical.

## 4. Materials and Methods

### 4.1. Experimental Model and Study Participant Details

All human IVD tissues used in the experiment were collected from trauma surgery patients with no history of back pain or IVD degeneration (donor characteristics are summarized in Table 1). Informed consent was obtained from the patients or their relatives, alongside general ethical approval from the University Hospital of Bern, Bern, Switzerland.

### 4.2. Human CEP Cell Isolation and Expansion

Within 24 h after surgery, tissue pieces were morphologically separated by NP, AF, and CEP. Human CEP cells were isolated from these CEP samples as previously described [18]. In brief, tissue was digested for 1 h in 1.9 mg/mL pronase (7 U/mg) (#10165921001, Roche Diagnostics, Basel, Switzerland) followed by overnight digestion in collagenase II ((285 U/mg) Worthington, London, UK) in low-glucose (1 g/L) Dulbecco’s Modified Eagle Medium (LG-DMEM; #31600083; Gibco, Basel, Switzerland) containing 10% *v*/*v* heat-inactivated fetal calf serum (FCS) (#F7524; Sigma-Aldrich, Buchs, Switzerland), 1% *v*/*v* penicillin/streptomycin (P/S), and 0.2% *v*/*v* primocin (#ant-pm; InvivoGen, San Diego, CA, USA, distributed by Lubioscience, Inc., Lucerne, Switzerland) on an orbital shaker at 37 °C. Following digestion, residual tissue was filtered through a 100 μm cell strainer, and cell quantification and viability were determined using trypan blue. Cells were then expanded in monolayer culture until passage two or three using high-glucose (4.5 g/L) Dulbecco’s Modified Eagle Medium (HG-DMEM; #52100039; Gibco, Basel, Switzerland) supplemented with 10% *v*/*v* heat-inactivated FCS and 1% *v*/*v* P/S. Cultures were maintained at 37 °C in a humidified atmosphere with 5% CO_2_ under normoxia, with medium changes twice per week.

### 4.3. Hydrogel Fabrication and Culture

Blended 2% agarose and 2 mg/mL type I collagen (Advanced BioMatrix, San Diego, CA, USA; #5133, 10 mg/mL) hydrogels were prepared as previously described by Cambria et al. [19]. In brief, 4% wt/vol agarose (2 g agarose powder; 50101, SeaPlaque low-gelling temperature, Lonza inc., Visp, Switzerland) was dissolved in 50 mL of 1× PBS, autoclaved at 120 °C, and maintained at 60 °C until use. In parallel, 4 mg/mL collagen solution was prepared by diluting bovine collagen type I (10 mg/mL; 5133, Advanced BioMatrix) with UltraPure Distilled Water according to the manufacturer’s instructions. Human CEP cells were trypsinized, then resuspended at a density of 15 × 10^6^ cells/mL in the collagen solution. The cell–collagen solution was then mixed 1:1 with the 4% wt/vol agarose solution to achieve a final concentration of 7.5 × 10^6^ cells/mL in agarose 2% wt/vol and 2 mg/mL collagen. Aliquots of 85 μL were pipetted into silicone molds (Ø = 6 mm, height = 3 mm) and allowed to set at room temperature (RT) for 10 min. Hydrogels were initially cultured in HG-DMEM with 10% FBS for six days with gradual serum deprivation for phenotype recovery and PCM formation [22,35,36].

After phenotype recovery, the blended collagen–agarose hydrogels were transferred to custom-made unconfined chambers and maintained in serum-free HG-DMEM supplemented with 1× ITS+ (1% Insulin, Transferrin, Selenium, BSA, and linoleic acid, I2521, Sigma, Buchs, Switzerland), 1× non-essential amino acids (11140-035, Gibco, Basel, Switzerland), 50 ug/mL (172 μM) L-ascorbic acid 2-phosphate (A8960, Sigma), and 10^−7^ M dexamethasone (D4902, Sigma) [37]. TNF-stimulated hydrogels were treated continuously with 10 ng/mL of TNF (#300-01A, Peprotech, London, UK).

Dynamic compression was applied using a custom-made bioreactor (Berner Fachochschule, Burgdorf, Switzerland) [38] delivering up to a 7% strain at 1.5 Hz for one hour, five days per week, for a maximum of two weeks as previously described [18]. As a control, unloaded hydrogels were cultured in the same custom-made chambers but did not undergo dynamic compression. Hydrogels were harvested at day 7 or day 14 immediately after the final loading cycle for downstream analyses, including cell viability, GAG and DNA quantification, and qPCR.

### 4.4. Stress–Strain Measurement

Hydrogels were prepared as described above; however, without the addition of cells. In addition to the blended 2% agarose and 2 mg/mL type I collagen hydrogels, 2% agarose hydrogels were prepared. To prepare 2% agarose hydrogels, a 1:1 mixture of 4% agarose and 1× PBS was pipetted into silicon molds (Ø: 6 mm, height: 3 mm), followed by incubation at RT for 15 min. All hydrogels were incubated at RT in PBS for 1 h to allow complete gelation.

Stiffness was determined using a modified protocol established by Frauchiger et al. [39]. Briefly, the bioreactor was used to impose a strain that increased by 3.33% every minute, up to a 30% strain. The Young’s modulus was taken by calculating the stress–strain ratio for the initial elastic region. The experiments were performed under unconfined compression at 37 °C in PBS to prevent dehydration of hydrogels.

### 4.5. Height Measurement

The heights of the hydrogels were measured at day 0, day 7, and day 14. Images of the carriers in each chamber were recorded prior to processing for downstream analysis. Heights of each hydrogel were measured using Fiji and averaged per each chamber.

### 4.6. Cell Viability

Cell viability of human CEP cells within collagen–agarose hydrogels was assessed using calcein-AM (#17783-1MG; Sigma-Aldrich, Buchs, Switzerland) and ethidium homodimer (#46043-1MG-F; Sigma-Aldrich) as live and dead cell markers, respectively. As previously described [18], on days 0, 7, and 14 timepoints, hydrogels were incubated in serum-free HG-DMEM supplemented with 5 µM calcein-AM and 1 µM ethidium homodimer for 35 min. at 37 °C. Following staining, samples were washed with PBS, and both the central and peripheral regions of each hydrogel were imaged with a confocal laser scanning microscope (cLSM710; Carl Zeiss, Jena, Germany). For quantification, the maximum number of live or dead cells was determined for each slice stack. Subsequently, slices located between the first and last slices containing over half of the maximum total cell number were analyzed using a custom macro developed for Fiji [40].

### 4.7. D Cell Morphology Imaging

To evaluate adhesion and cytoskeletal morphology in the collagen–agarose hydrogels, F-actin filaments and cell nuclei were measured on days 0 and 14 using Alexa Fluor 488-conjugated phalloidin and 4′,6-diamidino-2-phenylindole (DAPI), respectively [26]. Hydrogels were fixed with 4% formaldehyde overnight at 4 °C and then washed and stored in PBS until staining. The hydrogels were first permeabilized using 0.1% *v*/*v* Triton in PBS for 15 min at RT. After washing them 3× with PBS, hydrogels were incubated in the dark at RT for 2 h in 165 nM phalloidin in PBS with 1% BSA. Following staining, hydrogels were washed 3× with PBS and then incubated in the dark at RT for 20 min in 1 μg/mL DAPI diluted in 1× PBS. Finally, the samples were washed, and 3D stacked images were captured using a confocal laser scanning microscope (cLSM710; Carl Zeiss; Jena, Germany) with 10× and 20× dry objective lenses. Fiji was used for image processing to obtain a maximum intensity projection for each z-stack.

### 4.8. Cell Metabolic Activity

Metabolic activity of human CEP cells in collagen–agarose hydrogels was assessed on days 0, 7, and 14 using the Alamar blue assay, as previously described [18]. Hydrogels were incubated in 50 μM resazurin sodium salt solution in HG DMEM for three hours at 37 °C. The optical fluorescence index was measured with an ELISA plate reader (Spectramax M5, Molecular Devices, distributed by Bucher Biotec, Basel, Switzerland) at 544 nm excitation and 578 nm emission. Readouts were normalized to DNA content prior to group comparisons. Following measurement, hydrogels were washed and digested overnight at 60 °C in 0.625 mL papain solution, 3.9 U/mL papain (Sigma-Aldrich; #P-3125) and 5 mM L-cysteine hydrochloride (Sigma-Aldrich; #1161509). Digested samples were stored at −20 °C until used for determination of DNA and GAG content.

### 4.9. GAG and DNA Quantification

GAG content in the papain-digested samples was determined using 1,9-dimethyl-methylene blue (Sigma-Aldrich; #341088) as previously described [18]. Absorbance was read at 600 nm, quantified against a standard curve prepared with chondroitin sulphate (Sigma-Aldrich; #C6737).

DNA content from the same digested samples was measured using Hoechst 33,258 dye (#86d1405; Sigma-Aldrich) as previously described [18]. In brief, the optical fluorescence index was measured at 350 nm excitation and 450 nm emission, with values interpolated using a standard curve generated using calf thymus DNA sodium salt (#D1501; Sigma-Aldrich).

### 4.10. RNA Extraction and qPCR

On days 0, 7, and 14, hydrogels were snap-frozen in liquid nitrogen for gene expression analysis. RNA isolation was performed as previously described by Bougault et al. [22]. Briefly, hydrogels were solubilized in 1 mL of buffer consisting of 1.5 mL QG buffer (19063, Qiagen, Basel, Switzerland), 2 mL RLT lysis buffer (79216, Qiagen), 20 μL 2-mercaptoethanol, and RNA was extracted with the RNeasy Mini Kit (74106, Qiagen) and eluted in 30 μL RNase-free water. On-column DNase I (Sigma-Aldrich; #DNASE70-1SET) was used to remove genomic DNA.

RNA was reverse-transcribed to complimentary DNA (cDNA) using the High-Capacity cDNA kit (Thermo Fisher Scientific, Basel, Switzerland; #4368814) with a MyCycler™ Thermal Cycler system (Bio-Rad, Cressier, Switzerland; #1709703). qPCR was performed using iTaq Universal SYBR Green Supermix (Bio-Rad; #1725122) and gene-specific primers (Table 2) on a CFX96TM Real-Time System (Bio-Rad; #185-5096). Relative gene expression was calculated with the 2^−ΔΔCt^ method and normalized to the 18S reference gene and corresponding initial day 0 value.

### 4.11. Nitric Oxide (NO) Content

A Griess reaction assay was performed to quantify the NO radicals released by the hydrogels during culture as previously described [26]. In brief, the supernatant was mixed 1:1 with a solution of N-(1-naphthyl)ethylenediamine dihydrochloride (#N9125; Sigma-Aldrich), sulfanilamide (#S9251; Sigma-Aldrich), and 10% phosphoric acid (#79617; Sigma-Aldrich) and incubated at RT in the dark for 10 min. Subsequently, the absorbance was read at 530 nm.

### 4.12. Finite Element Model

A 2D axisymmetric porohyperelastic, compressible, Neo-Hookean FE model of a blended collagen–agarose hydrogel was developed in Abaqus 2024 using experimental and literature-derived material properties to investigate mechanical forces within the hydrogel (Table 3). The literature-derived permeability, Poisson’s ratio, and porosity were taken from studies of 2–3% agarose hydrogels [41,42] as this data was not available for blended collagen–agarose hydrogels. Young’s modulus was measured in the lab as described above (Section 4.4. Stress–Strain Measurement). Boundary conditions were implemented to simulate unconfined axial compression (Figure 6). In brief, the bottom surface was fixed in the vertical direction but free to move radially. The pore pressure of the lateral surface was set to zero to allow free fluid influx and outflux. Sinusoidal dynamic compression, with a displacement to 0.21 mm at 1.5 Hz, was applied to the top surface. Three different hydrogel heights (3 mm, 2.5 mm, 2 mm) were evaluated to assess height changes measured over the duration of the experiment. Simulations were run for 60 s.

RVF and RVT were monitored throughout the simulation at the nodes on the lateral surface of each model. The sum of RVF and of RVT across these nodes provided the net instantaneous flow rate and total cumulative fluid exchanged, respectively. These values provide insight into fluid absorption and expulsion, reflecting the interaction between the hydrogels and the surrounding fluids during the experiment.

### 4.13. Quantification and Statistical Analysis

All statistical analyses were performed assuming a non-parametric distribution. Specifically, results were evaluated using a Kruskal–Wallis test followed by a Dunn’s multiple comparisons post hoc test with a Benjamini–Hochberg correction to account for multiple comparisons across the factors of treatment, loading, and time. Gene expression, GAG, and cell metabolic data were normalized to the respective day 0 results and log-transformed before statistical analysis. All statistical analyses were performed using R (R Core Team, 2020) and RStudio (RStudio Team, 2020), specifically the R stats and dunn.test packages [43]. A *p*-value < 0.05 was considered statistically significant. All quantitative results are presented as a median up to six biological replicates, and the exact number of biological replicates (n) is indicated in each figure legend.

## Figures and Tables

**Figure 1 gels-11-00736-f001:**
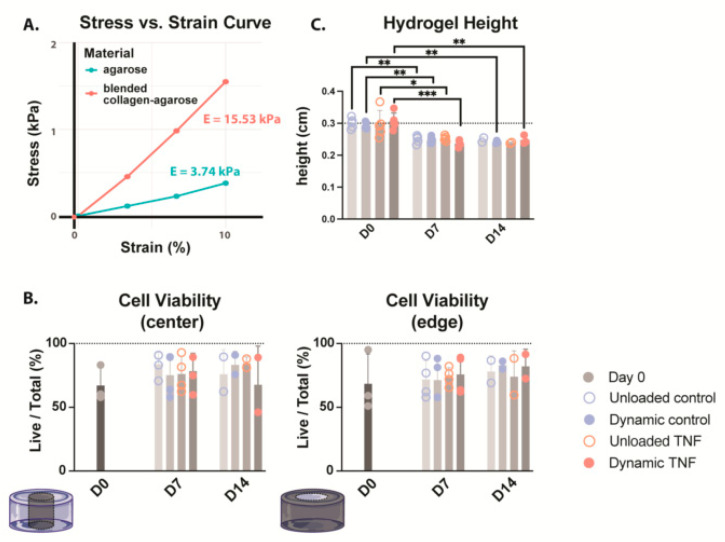
(**A**) Stress–strain curve comparing the stiffness of both 2% agarose (green) and blended 2% agarose and 2 mg/mL type I collagen hydrogels. Young’s modulus (E) is calculated as the slope of the stress–strain curve. *n* = 1 (**B**) Cell viability of CEP-seeded blended collagen–agarose hydrogels at the center and edge of the hydrogel. (**C**) Heights of the hydrogels for the duration of culture. * *p* < 0.05, ** *p* < 0.01, *** *p* < 0.001; *n* = 3–4.

**Figure 2 gels-11-00736-f002:**
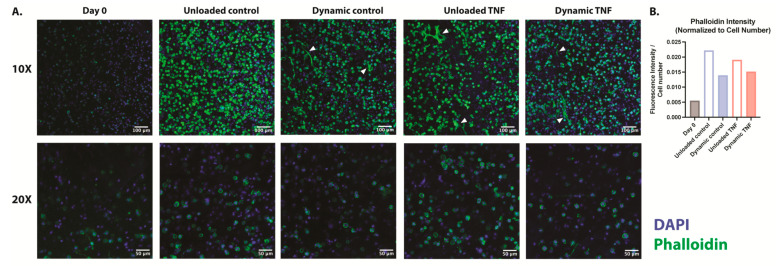
(**A**) F-actin staining with phalloidin of human CEP cells in blended collagen–agarose hydrogels at 10× and 20×. Phalloidin is shown in green, and DAPI is shown in blue. White arrows point to elongated cells. (**B**) Fluorescence intensity of phalloidin relative to cell number as determined by DAPI. *n* = 1.

**Figure 3 gels-11-00736-f003:**
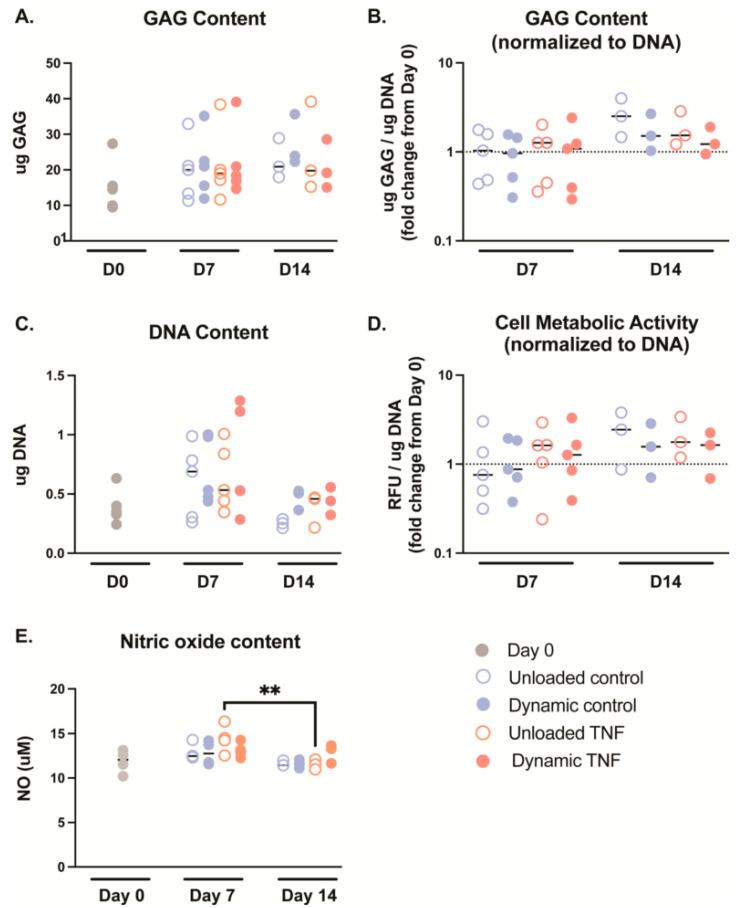
(**A**) Glycosaminoglycan (GAG) content measured in blended collagen–agarose hydrogels during culture. (**B**) GAG content normalized to the corresponding DNA content. (**C**) DNA content measured in blended collagen–agarose hydrogels during culture. (**D**) The cell metabolic activity of each condition was normalized to the corresponding DNA content. (**E**) The nitric oxide (NO) content released into the media on day 0 and after culture, *n* = 3–5. ** *p* < 0.01.

**Figure 4 gels-11-00736-f004:**
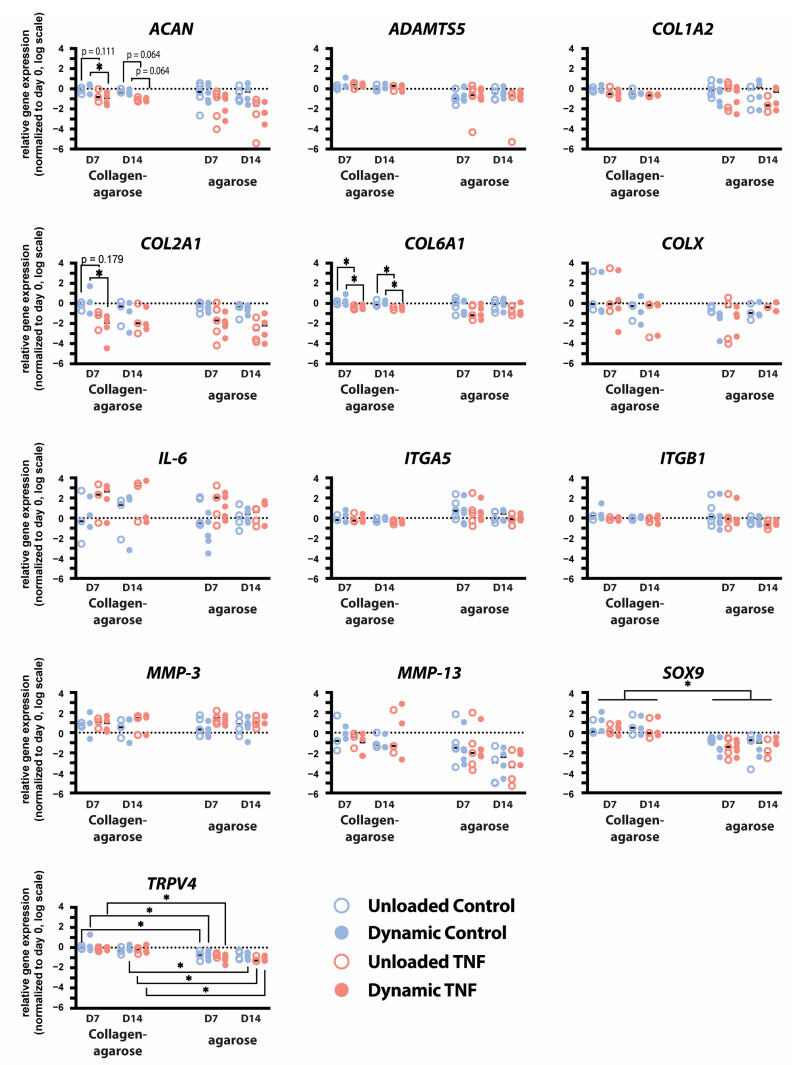
Relative gene expression of *ACAN*, *ADAMTS5*, *COL1A2*, *COL2A1*, *COL6A1*, *COL10A1*, *IL-6*, *ITGA5*, *ITGB1*, *MMP-3*, *MMP-13*, *SOX9*, and *TRPV4* in human CEP cells cultured in either blended collagen–agarose hydrogels or in 2% agarose hydrogels for 7 and 14 days. Data is normalized to *18S* and the corresponding day 0 expression. * *p* < 0.05; *n* = 3–5.

**Figure 5 gels-11-00736-f005:**
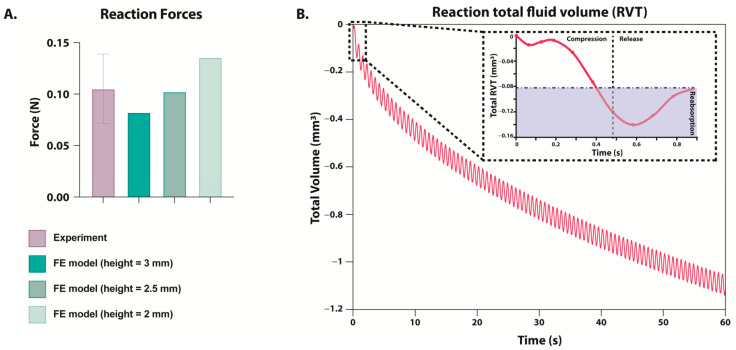
(**A**) Maximum reaction force during dynamic compression as measured experimentally (purple) or predicted in a FE model (green). FE model’s hydrogel height was simulated between 2 and 3 mm, but displacement remained at 0.21 mm. (**B**) Predicted reaction total fluid volume (RVT) measured during one minute of loading within the FE model. FE model’s hydrogel starting height was simulated to be 3 mm. The box in the top right corner zooms in on the change during one cycle of compression and release, and the area in blue is the fluid that has been reabsorbed before the next cycle of compression begins.

**Figure 6 gels-11-00736-f006:**
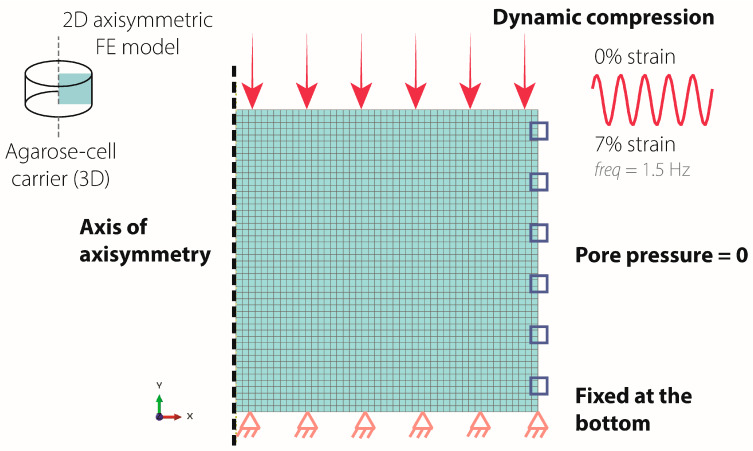
Boundary conditions of the axisymmetric FE model. The bottom surface was fixed in the vertical direction (orange triangle), while the lateral surface pore pressure was set to 0 (blue square). Sinusoidal dynamic compression of 0.21 mm displacement was applied to the top surface (pink arrow).

**Table 1 gels-11-00736-t001:** Donor characteristics summary.

Donor #	Gender	Age (Years)	Location
1 *	Female	42	L2/L3
2 *	Male	35	L4/L5
3	Male	28	L3/L4
4	Male	32	Th11/12
5	Female	22	Th11/12

* only used for day 0 and day 7 analysis due to limited number of cells following expansion.

**Table 2 gels-11-00736-t002:** qPCR primers.

Gene Type	Gene (Gene Abbreviation)	Gene ID	Forward and Reverse Primer Sequence
Reference Gene	18S	*18S*	f- CGA TGC GGC GGC GTT ATT Cr- TCT GTC AAT CCT GTC CGT GTC C
Anabolic Markers	Aggrecan	*ACAN*	f- CAT CAC TGC AGC TGT CACr- AGC AGC ACT ACC TCC TTC
	Type Icollagen	*COL1A2*	f- GTG GCA GTG ATG GAA GTGr- CAC CAG TAA GGC CGT TTG
	Type II collagen	*COL2A1*	f- AGC AGC AAG AGC AAG GAG AAr- GTA GGA AGG TCA TCT GGA
	Type VI collagen	*COL6A1*	f- TTCAAGGAGGCTGTCAAGAACr- TGATGAGGCGGTCGTAGG
	SRY-Box Transcription Factor 9	*SOX9*	f- GAG ACT TCT GAA CGA GAGr- GCT CTG ATG TGT TGA AGA AC
Catabolic Markers	A disintegrin and metalloproteinase with thrombospondin motifs 5	*ADAMTS5*	f- GCT GTG CTG TGA TTG AAG Ar- TGC TGG TAA GGA TGG AAG A
	Matrix metalloproteinase-3	*MMP-3*	f- CAA GGC ATA GAG ACA ACA TAG Ar- GCA CAG CAA CAG TAG GAT
	Interleukin 6	*IL6*	f- GCC ACT CAC CTC TTC AGA ACr- GCA AGT CTC CTC ATT GAA TCC A
	Type X collagen	*COLX*	f- GAA TGC CTG TGT CTG CTTr- TCA TAA TGC TGT TGC CTG TTA
Mechanoreceptors	Transient receptor potential cation channel subfamily V member 4	*TRPV4*	f- GTT GGT CTG GTC CTC ATT Gr- GAT TCC TGC TCG TCT ACT TG
	Integrin Subunit Alpha 5	*ITGA5*	f- ATC GCT CTC AAC TTC TCC TTr- CGG CTC TTG CTC TGA TAA TG
	Integrin Subunit Beta 1	*ITGB1*	f- CCT TGG TGT CTG TGC TGAr- GTC GTC AAC ATC CTT CTC CTT AC

**Table 3 gels-11-00736-t003:** FE model material properties.

Material Property	Value	Source
Permeability	6.61 × 10^−13^ m^4^/Ns	Gu et al. (2003) [43]
Poisson’s ratio	0.3	Tasci et al. (2014) [42]
Young’s modulus	15.53 kPa	-
Porosity	80%	Tasci et al. (2014) [42]

## Data Availability

The raw data supporting the conclusions of this article will be made available by the authors on request.

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
