# Peer review of "Exploring Mechanotransduction and Inflammation in Human Cartilaginous Endplate Cells in Blended Collagen–Agarose Hydrogels Under Cyclic Compression"

_gels, 2025, doi:10.3390/gels11090736_

Round 1

Reviewer 1 Report

Comments and Suggestions for Authors

  1. The abstract of the manuscript is very weak and does not reflect the true content of the current study. The abstract should be rewritten with the addition of the numbers of the results obtained.
  2. Line 56: There are incorrect phrases.
  3. There is no clear objective for this study. Authors should include a clear and explicit objective at the end of the introduction.
  4. Figure 1a The y-axis does not fit the result. The result should be redrawn and clarified to match the y-axis units.
  5. Figure 2a is not clear, and the images are silent and do not indicate anything to the reader. The shapes should be clearer to the readers.
  6. Figure 4 shows unclear axes and numbers. It is better to use logarithms on the y-axis to make it clearer for the reader.
  7. Conclusions should be free of results and numbers. No result may be added, as this is a chapter dedicated to the conclusion and not the results. Sources should also be deleted, as the conclusion is specific to the current study and not previous studies.
  8. Some working methods lack scientific references, making it difficult to go back and read the original or correct method. A reference must be added for each method.
  9. In the statistical analysis of the manuscript, the authors used a p-value < 0.05. This criterion is specific to agricultural and field experiments and should not be used in laboratory or medical experiments. The statistical analysis should be repeated at a p-value < 0.01 or 0.001.

Author Response

1. The abstract of the manuscript is very weak and does not reflect the true content of the current study. The abstract should be rewritten with the addition of the numbers of the results obtained.

The abstract has been rewritten to address your concerns.

2. Line 56: There are incorrect phrases.

This line has been deleted.

3. There is no clear objective for this study. Authors should include a clear and explicit objective at the end of the introduction.

We thank the reviewer for their feedback. A statement clearly stating the objective was added to the end of the introduction on lines 102-106.

4. Figure 1a The y-axis does not fit the result. The result should be redrawn and clarified to match the y-axis units.

We thank the reviewer for their feedback, however the values reported are correct. The Young’s modulus written on the graph is calculated the slope of of the respective stress-strain curve. We have added additional lines on the graph and clarification in the caption to address any possible confusion.

5. Figure 2a is not clear, and the images are silent and do not indicate anything to the reader. The shapes should be clearer to the readers.

We thank the reviewer for their feedback. The images are meant to show that a majority of the chondrocytes have retained their rounded shape. The quality of the image appears reduced in the submission and a higher quality image has been submitted for publication.

6. Figure 4 shows unclear axes and numbers. It is better to use logarithms on the y-axis to make it clearer for the reader.

The axes of Figure 4 have been updated to be more clear and labeled on a logarithmic scale.

7. Conclusions should be free of results and numbers. No result may be added, as this is a chapter dedicated to the conclusion and not the results. Sources should also be deleted, as the conclusion is specific to the current study and not previous studies.

We thank the reviewer for their comments on the Conclusions section. Only a summary of the results are included and the numbers are necessary to specify the hydrogel characteristics (2% w/v agarose and 2 mg/mL type I collagen). The statements needing references were removed. Additionally, we have changed the title of the section to “Conclusions and Limitations” because we also address limitations of the study.

8. Some working methods lack scientific references, making it difficult to go back and read the original or correct method. A reference must be added for each method.

References have been added to sections 4.2 and 4.5. Other sections within the methods, 4.3 and 4.9, are describing methods which were developed for this study and thus no relevant reference exists. Section 4.1 solely describes the access to samples and thus should not require a reference. All other methods sections have references.

9. In the statistical analysis of the manuscript, the authors used a p-value < 0.05. This criterion is specific to agricultural and field experiments and should not be used in laboratory or medical experiments. The statistical analysis should be repeated at a p-value < 0.01 or 0.001.

We thank the reviewer for their comment regarding the choice of statistical significance threshold. While we appreciate that certain fields may adopt more stringent criteria (e.g., p < 0.01 or p < 0.001), the use of p < 0.05 as a threshold for statistical significance is widely accepted in biomedical and laboratory-based research, including in the journal Gels. This convention originates from Fisher’s work in the early 20th century and remains standard practice across a broad range of experimental disciplines. In our manuscript, we report exact p-values for all statistical comparisons in the Results section, and in the figures we indicate significance with * (p < 0.05), ** (p < 0.01), and *** (p < 0.001). We therefore believe that our statistical reporting is both transparent and consistent with prevailing norms in biomedical research, while allowing readers to interpret the strength of evidence for each result.

Reviewer 2 Report

Comments and Suggestions for Authors

Compression parameters (7% strain, 1 Hz, 1 hr/day) may be sub-physiological for CEP cells. Prior studies suggest ≥10–20% strain for chondrocyte mechanoactivation. No justification for selecting 7% strain was provided. 

Despite 4× higher stiffness (Figure 1A), hydrogel modulus (15.53 kPa) remains far below native CEP (∼0.5–20 MPa). Calcified CEP in degeneration likely demands stiffer models. 

Height loss (~0.5 mm by Day 7, Figure 1C) was attributed to fluid expulsion (FE model, Figure 5B) but not experimentally quantified.

Background on hydrogel design should integrate recent advances in biomimetic matrices (e.g., 10.1016/j.celbio.2025.100020; 10.1016/j.bioactmat.2024.11.029).

F-actin staining (Figure 2A) revealed elongated fibers in TNF/dynamically loaded groups, suggesting fibrosis. Claims of "rounded morphology" were overstated given visible cytoskeletal changes. 

Permeability/porosity parameters (Table 3) were sourced from agarose (not collagen-agarose blends), reducing model validity. Predicted fluid loss (1.1 mm³/min) was not experimentally verified. 

Author Response

Reviewer 2

Compression parameters (7% strain, 1 Hz, 1 hr/day) may be sub-physiological for CEP cells. Prior studies suggest ≥10–20% strain for chondrocyte mechanoactivation. No justification for selecting 7% strain was provided. 

 We thank the reviewer for their feedback. 7% strain was chosen to match the study design in Crump et al. (2025) performing a similar experiment with CEP cell seeded into agarose. In order to compare the results of how a blended collagen-agarose hydrogel compares to 2% agarose alone, we kept the dynamic compression protocol the same. In the discussion, we recommend that higher loads would be necessary to induce chondrocyte mechanoactivation, and we have additionally added a sentence to the discussion on line 322-324 further justifying our choice.

Despite 4× higher stiffness (Figure 1A), hydrogel modulus (15.53 kPa) remains far below native CEP (∼0.5–20 MPa). Calcified CEP in degeneration likely demands stiffer models. 

We thank the reviewer for this observation. We note in the discussion on lines 305-309 that the stiffness does not reach that of native CEP tissue, however that it is close to the reported stiffness of the pericellular matrix of a chondron (17-200 kPa). This lower stiffness of blended collagen-agarose hydrogel is not an ideal representation of the CEP; however, it is an improvement from 2% agarose as we were comparing to in the previous study. We have added an additional sentence in the discussion on line 309 as a recommendation for future studies to address your concern.

Height loss (~0.5 mm by Day 7, Figure 1C) was attributed to fluid expulsion (FE model, Figure 5B) but not experimentally quantified.

We thank the reviewer for this observation. The contribution of fluid expulsion as a possible cause for height loss is solely based on predictions of the FE model, and the results and discussion have been reworded to clarify that these are only predictions which offer a hypothesis for the height loss seen. The reviewer is correct that this would need to be validated experimentally to confirm fluid loss as a cause for hydrogel height loss.

Background on hydrogel design should integrate recent advances in biomimetic matrices (e.g., 10.1016/j.celbio.2025.100020; 10.1016/j.bioactmat.2024.11.029).

 We thank the reviewer for their suggestion and have added additional information to the introduction on lines 81-91 regarding biomimetic hydrogels in cartilaginous tissues.

F-actin staining (Figure 2A) revealed elongated fibers in TNF/dynamically loaded groups, suggesting fibrosis. Claims of "rounded morphology" were overstated given visible cytoskeletal changes. 

We thank the reviewer for pointing out this overstatement. The results were updated to state that only the unloaded control hydrogels maintained a rounded morphology. For the other conditions, it was clarified that a majority of cells maintained the rounded morphology, but also showed elongated fibers.

Permeability/porosity parameters (Table 3) were sourced from agarose (not collagen-agarose blends), reducing model validity. Predicted fluid loss (1.1 mm³/min) was not experimentally verified. 

The reviewer is correct that ideally these values would come from blended collagen-agarose hydrogels, however this information was not available at the time of publication. Thus, studies of 2-3% agarose were used instead. This is described in the methods and has been added to the Conclusions and Limitations section.

Unfortunately predicted fluid loss from the FE model was not experimentally verified, as we did not have equipment in the lab to measure fluid loss experimentally during compression. The FE model is meant to offer insight into the mechanics of our hydrogels which we could not measure experimentally. Throughout the results and discussion, we make sure to state that fluid loss was only a prediction and not calculated experimentally.

Reviewer 3 Report

Comments and Suggestions for Authors

The authors have presented a well-written manuscript titled “Exploring Mechanotransduction and Inflammation in Human Cartilaginous Endplate Cells in Blended Collagen-Agarose Hydrogels Under Cyclic Compression”, which addresses a significant topic in tissue engineering and mechanobiology. However, several revisions are needed to improve the clarity, scientific rigor, and overall quality of the manuscript before it can be considered for publication. The following comments and suggestions are provided for improvement:

  1. Improve the abstract by clearly stating the novelty and uniqueness of the work.
  2. Update the introduction with recent literature and highlight the advancement over existing studies.
  3. Justify the absence of ROS assays, given their relevance to inflammation and macrophage activity.
  4. Explain the swelling/deswelling behavior of the hydrogel system.
  5. In Figure 2B, include visible error bars and clarify statistical significance.
  6. In Figure 5A, ensure error bars are shown or explain if triplicate tests were not performed.
  7. Discuss cell proliferation behavior in the hydrogel and justify biocompatibility claims.

Author Response

Reviewer 3

Comments and Suggestions for Authors

The authors have presented a well-written manuscript titled “Exploring Mechanotransduction and Inflammation in Human Cartilaginous Endplate Cells in Blended Collagen-Agarose Hydrogels Under Cyclic Compression”, which addresses a significant topic in tissue engineering and mechanobiology. However, several revisions are needed to improve the clarity, scientific rigor, and overall quality of the manuscript before it can be considered for publication. The following comments and suggestions are provided for improvement:

1. Improve the abstract by clearly stating the novelty and uniqueness of the work.

The abstract has been rewritten to address your concerns.

2. Update the introduction with recent literature and highlight the advancement over existing studies.

We thank the reviewers for their suggestion and have added additional information in the introduction as well as sentences clarifying the contributions of our studies to advance prior studies on the use of hydrogels for CEP culture.

3. Justify the absence of ROS assays, given their relevance to inflammation and macrophage activity.

We thank the reviewers for their suggestion to improve our analysis. To address this, we have performed a Griess reaction assay to measure nitric oxide secretion into the media. This data can be found in the Results (Section 2.4) and Methods (Section 4.9).

4. Explain the swelling/deswelling behavior of the hydrogel system.

Swelling was not directly measured in our study, however previous work by Cambria et al. with blended collagen-agarose hydrogels of the same composition (2% agarose and 2 mg/mL type I collagen) have been previously shown not to swell over time.

5. In Figure 2B, include visible error bars and clarify statistical significance.

Only an n=1 was done for phalloidin staining due to limited hydrogels available for downstream analysis. Thus, no statistics were run.

6. In Figure 5A, ensure error bars are shown or explain if triplicate tests were not performed.

In Figure 5A, error bars are only included for the experimentally measured reaction forces (shown in purple). The FE model only outputs one value for the maximum reaction force, thus no error bars are included.

7. Discuss cell proliferation behavior in the hydrogel and justify biocompatibility claims.

We thank the reviewer for their suggestion. Additional analysis regarding DNA content was added to the results (Section 2.3) regarding early cell proliferation after 7 days and the return to initial cell counts after 14 days. Additional sentences were added in the discussion on lines 267-271regarding biocompatibility.

Round 2

Reviewer 1 Report

Comments and Suggestions for Authors

After the second review of the manuscript, I see that the authors have made all the required corrections from the first review. I believe the manuscript is now clearer and ready for publication.